# A Systematic Review Protocol Investigating Community Gardening Impact Measures

**DOI:** 10.3390/ijerph16183430

**Published:** 2019-09-16

**Authors:** Jonathan Kingsley, Aisling Bailey, Nooshin Torabi, Pauline Zardo, Suzanne Mavoa, Tonia Gray, Danielle Tracey, Philip Pettitt, Nicholas Zajac, Emily Foenander

**Affiliations:** 1School of Health Sciences, Swinburne University of Technology, Melbourne, Victoria 3122, Australia; e.foenand@gmail.com; 2School of Social Sciences, Swinburne University of Technology, Melbourne, Victoria 3122, Australia; aabailey@swin.edu.au; 3School of Global, Urban and Social Studies, RMIT University, Melbourne, Victoria 3000, Australia; nooshin.torabi@rmit.edu.au; 4Department of Child Safety, Youth and Women, Queensland Government, Brisbane, Queensland 4000, Australia; Pauline.Zardo@csyw.qld.gov.au; 5Melbourne School of Population and Global Health, University of Melbourne, Melbourne, Victoria 3010, Australia; suzanne.mavoa@unimelb.edu.au; 6Centre for Educational Research, Western Sydney University, Sydney, New South Wales 2751, Australia; T.Gray@westernsydney.edu.au; 7Centre for Educational Research and Transitional Health Research Institutes, Western Sydney University, Sydney, New South Wales 2751, Australia; D.Tracey@westernsydney.edu.au; 8Botanic Gardens & Centennial Parklands, Sydney, New South Wales 2000, Australia; Philip.Pettitt@bgcp.nsw.gov.au; 9Faculty of Science, School of Environment, University of Auckland, Auckland, North Island 1010, New Zealand; nick.zajac@icloud.com

**Keywords:** community garden, systematic review, protocol

## Abstract

Existing community gardening research has tended to be exploratory and descriptive, utilising qualitative or mixed methodologies to explore and understand community garden participation. While research on community gardening attracts growing interest, the empirical rigour of measurement scales and embedded indicators has received comparatively less attention. Despite the extensive body of community gardening literature, a coherent narrative on valid, high quality approaches to the measurement of outcomes and impact across different cultural contexts is lacking and yet to be comprehensively examined. This is essential as cities are becoming hubs for cultural diversity. Systematic literature reviews that explore the multiple benefits of community gardening and other urban agriculture activities have been undertaken, however, a systematic review of the impact measures of community gardening is yet to be completed. This search protocol aims to address the following questions: (1) How are the health, wellbeing, social and environmental outcomes and impacts of community gardening measured? (2) What cultural diversity considerations have existing community garden measures taken into account? Demographic data will be collected along with clear domains/constructs of experiences, impacts and outcomes captured from previous literature to explore if evidence considers culturally heterogeneous and diverse populations. This will offer an understanding as to whether community gardening research is appropriately measuring this cross-cultural activity.

## 1. Introduction

Urbanisation and disconnection from the natural environment present significant public health, environmental and social challenges, due to their associations with increased social isolation, community health concerns, environmental degradation and poor mental health outcomes [1,2,3,4]. Community gardens have been viewed as a way of addressing and bridging these concerns by, for example, promoting improved community connections [1,5,6,7,8,9], enhancing ecological sustainability [10,11] and restoring psychological wellbeing [12,13]. Therefore, it is not surprising that there has been a growing appetite for establishing community gardens as preventative health, neighbourhood renewal and environmental policy strategies for “sustainable urban living” [1,7,11,14]. Other commentators have acknowledged that policy makers should consider developing community gardens because they are cost-effective [3], address food insecurities [15,16], promote fruit and vegetable intake [17,18,19], encourage contact with nature [10], support environmental regeneration and resilience [20], and enhance social capital [18,21,22,23]. To increase social capital and social cohesion, culture as a determinant of health has become an attractive policy response in urban areas across the world [24,25,26]. In addition, a growing body of psychological and architectural literature has reported positive associations between human experiences of connecting with the natural environment, and constructs such as wellbeing and vitality [27,28,29,30,31,32]. 

The health and environmental benefits of urban gardens are multi-faceted [33,34]. By providing green spaces that previously had not existed, community gardens enhance urban green infrastructure, while also providing new habitat for both fauna and flora [35]. This provides biodiversity benefits, and if strategically located within a network of other green spaces, urban community gardens can also act as a stepping stone for other species [36]. Community gardens soften the urban matrix that has resulted from roads and housing blocks fragmenting the landscape. This happens by reducing the rough patches like abandoned lands for living organisms [37]. Soil-based ecosystem services such as soil fertility and quality are another benefit provided by community gardens [38]. Climate regulation could be one of the benefits of community gardens in urban settings if the gardens are designed in a scale to contribute to the urban green infrastructure to provide such benefits [39]. 

Currently, community garden evidence is predominantly qualitative looking at the perceived ecological, health, wellbeing and social implications of this setting whereas quantitative methods applied in the literature are inconsistent [9,40,41]. This paper describes a systematic review protocol to explore the range of indicators and impact measures currently used in the community garden literature. The term ‘impact’ within this review refers to the immediate and direct implications of an action (which may include individual behaviour change or community support) and longer-term outcomes and consequences of a program (for example improved quality of life or increase in social capital in society) [42]. In the context of the present study, both qualitative and quantitative studies are equally valued, and thus usage of the word “impact” is not exclusively reserved to describe quantified causal inference, but also subjectively reported qualitative perceptions, understandings and narratives. Focusing on the measurement of community garden participation, this protocol recognises and builds on the work of Lovell et al. [3], who previously published a search protocol focused on the health and wellbeing impacts of community gardening for adults and children. Further, recent studies like Genter et al. [43], Ohly et al. [44], Kunpeuk et al. [45], and Audate et al. [40] have reviewed such benefits identified in the literature associated with allotment gardening, gardening, and urban agriculture accordingly. The following protocol paper is unique compared to these studies in that it investigates how data has been collected to facilitate more consistent, robust and valid community gardening information being gathered in the future.

### 1.1. Defining Community Gardening

Although substantial evidence highlights the benefits of community gardening, there is still variability in defining this activity [7]. At its most basic level, community gardening denotes a place for growing food (fruit, vegetables and livestock) and non-edible plants in a communal, collective, progressive and cooperative setting [15,41,46,47,48], whereas allotment gardening is a parcel of land acquired by individuals and/or family via a lease or rent for personal usage [43,49]. The proposed study uses the term “community gardening”, given this is the term most commonly used to describe communal, allotment and urban gardening on public land. With this in mind:
“Community gardening… now embraces a broad range of horticulture, environmental, social and political concerns… combining the “best of environmental ethics, social activism and personal expression” and involving “a faith that what they [gardeners] do not only helps the individual but strengthens the community””[50] (p. 945)

Community gardens allow a diverse population to come together to practice something they are mutually interested in, in a meaningful way to enhance social ties, leadership, strengthen community, and promote better health by deepening the reciprocity between neighbours [3,4,21,47,48,51,52,53]. As Crossan et al. [54] (p. 937) highlights:
“community gardens work can be generative of progressive forms of political practices that offer us glimpses of a radical future in the urban citizenry”

Challenges have been encountered when trying to measure the health, wellbeing, environmental and social experiences, outcomes and impacts of community gardening, as reflected in the myriad of studies which posit varying disciplinary approaches, with most based in the USA context [55]. These challenges concern inconsistent and different measurement approaches to explore the impacts of community gardening beyond the USA context. A more robust approach to community gardening impact measurement is required in order to consider community gardening as a public health and environmental policy initiative. Barnidge [17] (p. 7) supports this contention, explicitly recommending that, in relation to knowledge translation:
“further research is needed to develop systematic approach for scaling up [community garden] intervention”

This is specifically critical as most studies exploring cultural domains in the urban areas focus on the socio-demographic and migration while policy makers are seeking new ways to deal with governing culturally diverse cities [56]. It also cannot be assumed that community gardening outcomes are beneficial for all local residents living in neighbourhoods, as there may be a range of unintended consequences. Indicators and impact measures should be able to determine both positive and negative consequences, like the exclusion that can occur between community garden members and those in the community who are not members [52]. Talbot and Verrinder [42] provide an effective policy model to measure this, acknowledging that the first step in this process is to identify local community issues. In the context of community gardening, the model developed by Talbot and Verrinder would advocate exploring (i) epidemiological, (ii) demographic, (iii) qualitative descriptive and (iv) statistical data on the potential for community gardens to impact communities in terms of health, wellbeing, social, environmental, educational and cultural factors.

### 1.2. Objectives of the Review 

The objectives of this review are to identify existing quantitative and qualitative measures of community gardening, and to assess the empirical rigour of these measures. Therefore, a mixed methods/mixed studies review is applied. A tabulated summary of existing impact measures will be presented, noting (i) methodology (qualitative or quantitative), (ii) domains of measurement based on different disciplinary concepts (see Table 1), (iii) cultural validity information provided, and (iv) limitations. The following research questions will be used to frame the reporting of review results:

(1) How are the health, wellbeing, social and environmental experiences, impacts and outcomes of community gardening being measured? 

(a) What existing impact measures are being used to collect information on community gardening? 

(2) What cultural diversity considerations have existing community gardening impact measures taken into account? 

(a) Have any existing community gardening impact measures been validated in culturally diverse contexts?

By approaching the proposed review via the lens of the aforementioned research questions, it is the intention of the authors to establish a comprehensive understanding of the breadth and scope of existing approaches to community garden measurements.

## 2. Methods

The proposed approach is designed to capture and synthesise all existing peer-reviewed literature concerned with the qualitative or quantitative measurement of community gardening impacts at individual, community, and/or environmental levels. The review will apply the *Preferred reporting items for systematic reviews and meta-analyses* (PRISMA) search checklist, statement, protocol and diagram to inform this research project [57,58].

Data screening, and/or synthesis of articles authored by any members of the research team will be conducted by two members of the team who have not authored the article(s), to control for author bias. Members of the research team will participate in meetings throughout the review process in order to provide progress reports, share preliminary review findings, and invite constructive feedback. Such meetings will also provide a forum at which the team can begin to consider and plan the required key features of the measurement instrument this review is designed to inform. The findings of the systematic review will be formatted for publication, and the formal design of our proposed new measurement instrument will ensue.

### 2.1. Searches

The search scope will be inclusive of international literature that is published in English due to the scope, timeline and funding of this project. Preliminary scoping searches indicate that the earliest community gardening impact measurement publications in English emerged in 1993, and as such the search scope of this protocol is set to include all results published from 1993 to the present time. The preliminary scoping review established the suitability of this timeline. Although different approaches and technologies for scoping literature exist (for example, Citespace Software that track research hotspots), we deemed that this went beyond the scope of the proposed systematic review.

The transdisciplinary research team who will review these publications come from disciplines including public health (e.g., health promotion, epidemiology), sociology, anthropology, psychology, and environmental science (e.g., horticulture, biodiversity and geology). Search results from each disciplinary search string will then be returned to members of the research team to determine inclusion or exclusion based on the criteria stated in subsequent sections of this protocol. It is intended that this systematic approach will inform a comprehensive search of community gardening impact measurement literature published in English. Academic searches will be conducted using: (1) EBSCOhost, (2) Web of Science, (3) SOCIndex (EBSCOhost), (4) Sociology Database (ProQuest), (5) Social theory, (6) SCOPUS, (7) Academic search complete, (8) ScienceDirect(Elsevier), (9) CAB abstract, (10) CINHAL, (11) PsycINFO and (12) GeoRef. Supplementary data will include the authorship team’s existing community gardening research repositories, consolidated and managed in Endnote. 

This paper posits that the lack of methodological consensus and measurement precedent evident in existing approaches to community gardening research, can be responded to by: critically appraising existing measures, andproducing a review of the strengths and limitations of existing measurement approaches. 

Evidence on the cultural validity of existing impact measurement scales is yet to be synthesised and evaluated. This task is included within the scope of the proposed review. 

### 2.2. Search Strategy and Terms

As a response to the methodological problem of siloed community gardening impact measurements, the authors of this paper combine their expertise as a collective. In doing so, this protocol reflects a collaboratively formulated systematic review plan that mirrors transdisciplinary consensus. The list of search terms outlined in the Table 1 were drafted and refined following a thorough consultation process inclusive of the authors of the paper. This search strategy reflects the principles of co-design [59], inclusive and reflective of the diverse expertise and disciplinary perspectives of the team. Search terms were then formed into strings designed to capture relevant results from each authors’ disciplinary field. Search strings were refined based on tests conducted using the aforementioned databases. This process led to different combinations of keywords being used for disciplinary specific databases which can be seen as both a strength and weakness of the subsequent review. The strength of this approach is its ability to capture discipline-specific literature; however, this process may lead to some inconsistencies in the articles identified. 

Five search strings were developed using key terms from Table 1. The authors recognise that this is not an exhaustive list of terms, but align with the areas identified by the authorship group. Twelve different databases will be searched for academic literature results. At the time of writing, no existing search protocols, specifically focused on the impact measurement of community gardening (published in English) were locatable. As such, benchmarking of the search strategy proposed in this paper was not possible. However, the search strategy refinement process the authorship team have engaged in to inform the development of this protocol, has included an active commitment to developing a comprehensive search approach.

(i) Web of Science, CINHAL and EBSCOhost:

(“Community garden” OR “allotment garden*”) AND (health OR wellbeing OR “social capital” OR “social cohesion” OR “food security” OR education OR “determinant of health” OR safety) 

(ii) SOCIndex, CAB abstract and Sociology Database (ProQuest):

(“Community garden” OR “urban gardening) AND (urban agriculture OR natural environment OR “growing food” OR “sense of place”) AND (reciprocity OR connection OR wellbeing OR community) 

(iii) Social Theory and SCOPUS:

“Urban food garden” AND (“social equity” OR “social inclusion” OR “community engagement”) AND (“ecological benefits” OR biodiversity OR “urban green spaces”) AND (“place making OR “cultural landscape” OR resilience OR conservation OR governance)

(iv) Academic search complete, PsycInfo and ScienceDirect(Elsevier):

“community garden*” AND (“environmental psychology” OR ecopsycholog* OR psychological OR inclusion OR belonging) AND (measure OR scale OR outcome OR quant*) 

(v) GeoBase (Elsevier) and Web of Science:

“Community garden*” AND (“geological history” OR soil OR “ground water” OR mineral*) AND (nature OR sustainability) AND (education OR “environmental design” OR connection)

### 2.3. Article Screening and Study Inclusion Criteria

The relevant subject for this review is the measurement of “community gardening” experiences, impacts and outcomes. All results returned from the aforementioned searches will be considered for inclusion by the research team during an initial filtering stage. All studies identified as not relevant to the current task, due to not being focused on the measurement of community gardening, will be excluded from further review stages. Studies describing the use of community gardening as a community (or other) intervention will be included within the scope of this review. All qualitative and quantitative studies meeting the inclusion criteria will be included. The authors recognise that “community gardening” and “allotment gardening” are different activities, but have opted to include both forms of gardening, given the similarity and equal relevance in reference to an urban agriculture activity.

#### 2.3.1. Inclusion Criteria

The rationale for the mixed methods approach taken is to draw on the rich data and in-depth perspectives provided by both qualitative and quantitative scholars. Both qualitative and quantitative scholars have contributed to the field of community gardening research. It is also recognised that impact can be measured via qualitative methods. Given that the overarching aim of this systematic review is to synthesise the existing work in this space, and unite siloed narratives, it is essential to include both paradigms across disciplines. The authors have also included different age groups, gardening forms (e.g., school gardening) and different qualitative and quantitative study designs to gather rich data.

#### 2.3.2. Qualitative Studies 

Test searches suggest that the vast majority of search results this review will yield will be qualitative. All papers using qualitative methodology that describe the design or use of a measure/s to explore impacts of community and/or allotment gardening will be included in this review, as these studies can offer valuable insights and enable understanding about existing measurement precedents and limitations. 

#### 2.3.3. Quantitative Studies

Supplementary searches described in prior sections of this paper suggest that a limited number of quantitative community gardening measurement approaches exist. All quantitative papers that describe the design or usage of measures to explore community gardening impacts will be included in this review. 

#### 2.3.4. Exclusion Criteria

All studies not meeting the inclusion criteria will be excluded from this review. Grey literature will not be included in this review because it goes beyond the scope of the proposed research project. All literature published prior to 1993 will be excluded from this review to ensure that only relatively recent seminal work is included in the review process. Only original research will be included in the systematic review and therefore articles such as literature reviews or commentary articles will be excluded. A list of articles excluded from the review will be provided in tabulated form, accompanied by the rationale for exclusion.

### 2.4. Article Evaluation Approach

The review approach will be staged with one researcher extracting the literature from the academic databases. A secondary review provided by the lead author of this paper will involve checking for consistency of decisions, by reviewing a minimum of 10 percent of (i) titles, (ii) abstracts and (iii) full text results, for each search string and database combination mentioned in the previous section. As previously mentioned, further inclusion criteria include (i) studies published in English, and (ii) studies published from 1993 to the present time. Articles will then be allocated to respective authors for screening based on disciplinary alignment. These disciplinary leads will review these papers in full outlined below. 

Qualitative impact measurements will be evaluated based on the following lists of measurement criteria. Upon completion of data collection and screening, these criteria will be used to synthesise results and to inform the development of an overarching critique on the strengths and limitations of existing community garden measurement approaches. This process will involve reviewing articles that meet the inclusion criteria and critiquing the presence or absence of the features mentioned below.

Qualitative measurements will be evaluated via consideration of the following assessment criteria (Phillips, Street and Haesler [60] assisted in the development of this checklist): ▪What are the available qualitative measurements that can be used to measure community gardening experiences, outcomes and impacts?▪What criteria do available qualitative measures incorporate?▪Are the identified qualitative measurements used to measure community gardening experiences, outcomes and impacts methodologically sound?▪What is the feasibility of the existing approaches to measure community gardening experiences, outcomes and impacts in diverse socio-cultural contexts? 

Quantitative measurements will be evaluated via consideration of the following adapted criteria [49]:▪What are the available quantitative measurements that can be used to measure community gardening experiences, outcomes and impacts?▪What criteria do available quantitative measures incorporate?▪Are the identified quantitative measurements used to measure community gardening experiences, outcomes and impacts reliable and valid?▪What is the feasibility of the identified quantitative measurements to measure community gardening experiences, outcomes and impacts in diverse socio-cultural contexts? 

The overall quality of both the qualitative and quantitative papers will be assessed by applying Trisha Greenhalgh [61,62] hierarchy. A cultural dimension will be applied to these questions, by including sub questions to ensure that appropriate data relative to the scope of this review is retrieved. Measures used will be categorized thematically and descriptive statistics for each category reported accordingly.

### 2.5. Data Extraction Strategy

This study will not collect or report on outcome data. The focus of this study is on reviewing measurements and approaches exclusively. Only data pertaining to the methods used to measure community gardening impacts will be extracted, tabulated, and discussed in the review. The table of included studies will be cross-referenced to inform responses to the research questions. An Endnote repository of included (and excluded) search results will be consolidated and maintained by the first author of this paper. Academic literature that meets the inclusion criteria will be tabulated as mentioned in previous sections, and a stepped data extraction process will be described, to ensure the systematic review approach applied is replicable. Extracted data records will be made available as additional files. In any instance(s) where reporting of qualitative or quantitative research findings appear unclear or inconclusive, corresponding authors will be emailed and invited to clarify their findings for inclusion in the proposed review output. In any instances where clarification is not successfully obtained, inconclusive or unclear findings will be made salient.

## 3. Limitations

The authors recognise the limitation of not including authors/contributors from disciplines like urban design, economics and architecture, which for example may have theorised domains in Table 1 differently. Further, as four authors of this protocol have come from anthropology, research translation, community psychology and social science backgrounds, all of which require the researcher to practice reflexivity, it is critical to explore our own biases associated with this study based on a reflexive approach [63]. The research team acknowledges its pre-existing theoretical bias toward assuming that time spent in the natural environment is likely to be positively associated with improved human health, social and environmental outcomes. The research team will remain aware of this bias during data collection, and systematic review processes, and will maintain efforts to approach data with neutrality. A colleague not listed on the authorship team, with a public health background and no conflicting or vested interest in the trajectory or outcomes of this study, will review the systematic review to ensure the maintenance of an iterative process is incorporated [64].

## 4. Conclusions

Search results will be evaluated against the list of criteria stated in the “Article evaluation approach” section, to inform critical discussion of the strengths and limitations of existing approaches to community gardening impact measurement. A narrative synthesis of results will subsequently be presented. This will include commentary on the ways that different disciplines have approached community gardening impact measurement to date, evaluation of which impact domains different disciplines measure, and the ways that a transdisciplinary model can advance existing practice in community gardening impact measurement, by enabling dynamic and comprehensive assessment. It is anticipated that results from the proposed systematic review will develop a better understanding of measures for collecting health, wellbeing and ecological data to assess the impacts and outcomes of community gardening. Specifically, the authors of this paper aim to apply the findings of the systematic review to inform the development of an empirically validated measurement scale for community gardening. In a recent scoping review of the literature focused on community gardening and its wellbeing implications for vulnerable populations [65], the need to overcome inconsistent measures was seen as fundamental. With evidence [53,66,67,68,69] highlighting the potential of community and allotment gardening to support marginalised populations, such a review is critical to ensure this activity is available to all. The proceeding systematic review will fill this void in knowledge in an attempt to develop consistent measures of community gardening that cuts across different segments of the population.

## Figures and Tables

**Table 1 ijerph-16-03430-t001:** Transdisciplinary community garden search terms.

Domains	Public Health	Sociology and Environmental Anthropology	Biodiversity	Environmental and Community Psychology	Earth Science
Environmental terms	community garden allotment garden	community garden urban agriculture urban gardening natural environment	urban food garden urban green spaces ecological benefits biodiversity conservation	community garden environmental psychology ecopsychology	community garden geological history soil ground water mineral nature sustainability environmental design
Health terms	Health; wellbeing	Wellbeing		quality of life	
Social determinant terms	determinant of health social capital social cohesion education	sense of place	social inclusion social equity		Education
Community terms	safety	community reciprocity connection	community engagement place making cultural landscape resilience	Inclusion belonging	Connection
Food production terms	food security	growing food			
Measurement terms				Measure scale outcome quantitative	
Other			Governance

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
