# Peer review of "A Systematic Review Protocol Investigating Community Gardening Impact Measures"

_ijerph, 2019, doi:10.3390/ijerph16183430_

Round 1
Reviewer 1 Report
Dear Authors,
thank you for sharing your research protocol. This will surely support other research teams working on community gardening.
Here a few comments:
In the abstract the authors state that they seek to identify measures for engagement, outcomes and impact. However, the research questions only mention outcomes and impact? The authors should explain better what type of engagement their are referring to. Is it engagement during the design and development? Or do you associate engagement with use?
Statement on page 2 lines 74-76 should be supported by evidence.
Different combinations of key words are used for the different databases. The authors should explain why they decided to do so and how this will impact on their results.
Conclusion could be improved by discussing more in-depth the expected results and the potential contribution rather than presenting the research method again.
Author Response
Dear Reviewer 1,
Thank you for your time and feedback.
Please find my response to your feedback in the attached cover letter.
Take care,
Dr Jonathan (Yotti) Kingsley

Reviewer 2 Report
Research on community gardening attracts growing interest, the purpose of this paper is to use systematic literature reviews that explore the multiple benefits of community gardening and other urban agriculture activities to fill the void in knowledge and develop consistent measures of community gardening. The paper is of great significance. I believe this paper is very helpful to scientists. And I have only one question for the author's consideration.
Authors can use Citespace software (Citespce software is also beased on the database like EBSCOhost, Web of Science, SCOPUS, etc.) to track research hotspots (http://cluster.cis.drexel.edu/~cchen/citespace/), for some ideas and thoughts may proposed long ago, but refocused in recent yerars, or those the newest may not be the best. For example, in the draft, authors mention: All literature published prior to 1993 will be excluded from this review to ensure that only relatively recent seminal work is included in the review process. The authors fix the time to 1993 seems for no reason.Author Response
Dear Reviewer 2,
Thank you for your time and feedback.
Please find my response to your feedback in the attached cover letter.
Take care,
Dr Jonathan (Yotti) Kingsley
